# Loss of height predicts total and cardiovascular mortality: a cohort study of northern European women

Sofia Klingberg [1,2] Kirsten Mehlig,[1] Rojina Dangol,[3] Cecilia Björkelund [4] Berit Lilienthal Heitmann [3,5] Lauren Lissner[1]

For numbered affiliations see end of article.

**Correspondence to**
Sofia Klingberg;
sofia.klingberg@gu.se

## ABSTRACT

**Objective** To examine height changes in middle-aged northern European women in relation to overall and cardiovascular mortality.

**Design** Population-based cohort studies with longitudinally measured heights and register-based mortality.

**Setting** Sweden and Denmark.

**Participants** Population-based samples of 2406 Swedish and Danish women born on selected years in 1908–1952, recruited to baseline examinations at ages 30–60, and re-examined 10–13 years later.

**Main outcome measure** Total and cardiovascular disease (CVD) specific mortality during 17–19 years of follow-up after last height measure.

**Results** For each 1 cm height loss during 10–13 years, the HR (95% CI) for total mortality was 1.14 (1.05 to 1.23) in Swedish women and 1.21 (1.09 to 1.35) in Danish women, independent of key covariates. Low height and high leisure time physical activity at baseline were protective of height loss, independent of age. Considering total mortality, the HR for major height loss, defined as height loss greater than 2 cm, were 1.74 (1.32 to 2.29) in Swedish women and 1.80 (1.27 to 2.54) in Danish women. Pooled analyses indicated that height loss was monotonically associated with an increased mortality, confirming a significant effect above 2 cm height loss. For cause-specific mortality, major height loss was associated with a HR of 2.31 (1.09 to 4.87) for stroke mortality, 2.14 (1.47 to 3.12) for total CVD mortality and 1.71 (1.28 to 2.29) for mortality due to causes other than CVD.

**Conclusion** Height loss is a marker for excess mortality in northern European women. Specifically the hazard of CVD mortality is increased in women with height loss during middle age, and the results suggest that the strongest cause-specific endpoint may be stroke mortality. The present findings suggest attention to height loss in early and mid-adulthood to identify women at high risk of CVD, and that regular physical activity may prevent early onset height loss.

## INTRODUCTION

Adult height is generally maintained from the end of puberty until the beginning of the fifth decade at which time height starts to decline.[1–4] Height loss is a process caused by shrinking of vertebral discs,[5]

### Strengths and limitations of the study

► The population-based sampling, the prospective designs, the standardised measurement protocols and the long durations of follow-up for endpoints through high-quality national registries are major strengths of this study.

► The number of deaths due to stroke was low and results regarding stroke should therefore be interpreted with some caution.

► Based on the observational design we cannot rule out residual confounding by risk factors that were not measured in the study populations.

spinal compression fractures[6] and change in posture,[7] accelerating from the seventh decade of life.[1–4] Height loss is a predictor of both low bone mineral density[6] and clinical manifestations of osteoporotic fractures, which show a gradient across the northern-southern latitude.[8]

Although height loss could be thought of as part of the normal ageing process, a rapid decline has been suggested to predict overall mortality risk in two studies of men,[9 10] one in both sexes combined[11] and a single study specifically in women.[12] Height loss has also been associated with deaths due to cardiovascular disease (CVD) in men[10] and both sexes combined.[11] However, most studies have been performed in elderly populations and no study has to our knowledge reported sex-specific estimates for women whose height changes were measured in mid-adulthood. The fact that the effects of height loss in women has not been studied more thoroughly is remarkable because women tend to lose more height than men.[1–4] Thus, the aim of the present study was to determine if height loss in mid-life predicts overall mortality and mortality due to CVD in two longitudinal, population-based samples of middle-aged Nordic women.

## METHODS

### Study populations

Data from two prospective cohort studies were analysed: the Swedish Prospective Population Study of Women in Gothenburg (PPSWG) and women in the Danish arm of the MONItoring trends and determinants of CArdiovascular disease (MONICA) study. The analyses were performed in each cohort separately and after pooling of the cohorts.

The PPSWG, initiated in 1968–1969, recruited a sample of 1462 women, born in 1908, 1914, 1918, 1922 and 1930 on specific dates that were distributed evenly across birth years. Over 90% of the invited women participated at the baseline health examination.[13] In 1980–1981, a re-examination took place. A total of 1153 women were examined in PPSWG at both time points, and were thus eligible for the present study.

The Danish MONICA study recruited 1765 randomly selected female residents of western Copenhagen born in 1922, 1932, 1942 and 1952 to a health examination in 1982–1984.[14] The baseline participation rate was 79%.[14] A re-examination took place in 1994, in which 1264 women took part, constituting the eligible sample from MONICA.

The total number of Swedish and Danish women for this study was thus 2417. Exclusions of six Swedish women and five Danish women due to an implausible height increase of at least 2 cm left a final analytical sample of 2406 women.

### Height and other covariates

At both examinations, height was measured with a stadiometer to the nearest 0.5 cm on subjects without shoes. Measurement of height was generally performed early in the day. Height loss was calculated as baseline height minus height at follow-up. Thus, a positive value indicated height loss.

Potential confounders (age, time between measures, and baseline measures of height, weight, smoking, leisure time physical activity (LTPA), alcohol intake and education) were selected based on their known associations with height loss and mortality. All confounders were fully harmonised between the studies and included in fully adjusted models. Time between measures was calculated as the difference in time between follow-up measure and baseline measure. Participants were weighed wearing light clothing on a calibrated scale. Information on all other covariates was collected by questionnaires. Smoking status was categorised into never, former and current smokers. LTPA was assessed by a question placing the participants in one of four categories ranging from sedentary to vigorous activity.[15] Participants were then classified into three categories: almost inactive (low LTPA); at least 4 hours of low impact physical activity per week (medium LTPA); or regular physical exercise or competitive sports (high LTPA). Habitual intake of beer, wine and spirits during the last year was assessed based on which alcohol intake per day was estimated. Educational level was dichotomised into compulsory education versus more than compulsory. In PPSWG compulsory education ranged from 6 to 7 years in the different birth cohorts, while 7 years of schooling was compulsory for all birth cohorts in MONICA. Information on menopausal status was available but not included because of the collinearity with age.

### Mortality outcomes

Date and cause of death were assessed through national mortality registries. The Danish cohort was followed for total mortality until 4 October 2012 with a maximum follow-up of 19.3 years, and until 31 December 2010 for cause-specific mortality with a maximum follow-up of 16.7 years. The Swedish cohort was followed until 2014, but follow-up was restricted to 19.3 and 16.7 years for total mortality and cause-specific mortality, respectively, to harmonise with the Danish cohort. Death due to CVD was identified by International Classification of Diseases (ICD-8/9) codes 390–459 or ICD-10 codes I00–I99, and death specifically due to stroke was identified by ICD-8/9 codes 430–434 or ICD-10 codes I60–I64. Unspecified or uncertain stroke diagnoses in the Swedish cohort were verified through medical records.[16]

### Statistical analyses

The Cox proportional hazards model was used to investigate if height loss was associated with overall mortality and cause-specific mortality. Time of follow-up from last height measure until date of death or censoring was used as an underlying time metric. Height loss was parameterised as linear predictor and as binary predictor. Height loss was dichotomised at 2 cm, defining major height loss by height loss greater than 2 cm. Restricted cubic spline regression was used to further investigate non-linear associations between height loss and HR for the outcomes. Three knots were automatically placed at −0.7, 0.8 and 2.5 cm, with the reference set to zero (no height loss). Because of missing data for smoking, LTPA, education and/or alcohol intake, 74 Danish and 12 Swedish women were excluded from models that adjusted for these covariates. The proportional hazards assumption (PHA) of the Cox model was tested by inclusion of a product term between the major height loss variable and survival time until death or censoring for the respective outcome. These tests did not indicate violation of the PHA (all p>0.64). Linear regression and logistic regression were used for investigation of association between covariates and height loss.

To explore effect modification by baseline age, interactions between age (dichotomised at <50 vs ≥50 years) and height loss (linear and dichotomised) on overall mortality were investigated. To rule out influence of large height losses on the linear analyses, women with height loss ≥5 cm and ≥4 cm, respectively, were excluded in sensitivity analyses of the main outcome, total mortality. Analyses were also repeated after exclusion of deaths within the first 2 years after start of follow-up.

**Table 1** Characteristics of the two study populations and the pooled sample

| | Swedish women (n=1147) | Danish women (n=1259) | Pooled sample (n=2406) |
|---|---|---|---|
| Age distribution at baseline (years) | | | |
| 30–32 | – | 358 (28.4) | 358 (14.9) |
| 38–42 | 305 (26.6) | 344 (27.3) | 649 (27.0) |
| 46–47 | 332 (29.0) | – | 332 (13.8) |
| 50–52 | 323 (28.2) | 321 (25.5) | 644 (26.8) |
| 54–55 | 138 (12.0) | – | 138 (5.7) |
| 60–62 | 49 (4.3) | 236 (18.8) | 285 (11.9) |
| Age at baseline height examination (mean (SD), years) | 47.1 (6.1) | 44.3 (10.8) | 45.6 (9.0) |
| Age at last height examination (mean (SD), years) | 59.1 (6.1) | 55.2 (10.8) | 57.1 (9.1) |
| Height at baseline (mean (SD), cm) | 163.7 (5.8) | 163.7 (6.1) | 163.7 (6.0) |
| Weight baseline (mean (SD), kg) | 64.3 (10.3) | 63.1 (10.5) | 63.7 (10.4) |
| Height loss (mean (SD), cm) | 1.09 (1.21) | 0.6 (1.0) | 0.84 (1.12) |
| Time interval (mean (SD), years) | 12.0 (0.2) | 10.9 (0.3) | 11.4 (0.6) |
| Rate of height loss (mean (SD), cm/year) | 0.09 (0.10) | 0.06 (0.09) | 0.07 (0.10) |
| Education: compulsory or less at baseline | 793 (69.3) | 411 (32.6) | 1204 (50.1) |
| Smoking at baseline | | | |
| Never | 603 (52.6) | 415 (33.0) | 1018 (42.3) |
| Former | 89 (7.8) | 215 (17.1) | 304 (12.6) |
| Current | 454 (39.6) | 629 (50.0) | 1083 (45.0) |
| Leisure time physical activity at baseline | | | |
| Low (almost inactive) | 198 (17.3) | 367 (29.2) | 565 (23.5) |
| Medium (≥4 hours low impact leisure time physical activity/week) | 812 (71.0) | 718 (57.3) | 1530 (63.8) |
| High (regular physical activity or competitive sports) | 136 (11.9) | 174 (13.8) | 310 (12.9) |
| Leisure time physical activity at follow-up | | | |
| Low (almost inactive) | 335 (29.2) | 311 (26.1) | 646 (27.6) |
| Medium (≥4 hours low impact leisure time physical activity/ week) | 582 (51.0) | 735 (61.8) | 1317 (56.5) |
| High (regular physical activity or competitive sports) | 230 (20.1) | 146 (12.3) | 376 (16.1) |
| alcohol intake (mean (SD), g/day) | 8.3 (10.9) | 9.8 (11.6) | 9.1 (11.3) |

Data are numbers (%) unless indicated otherwise.

Interactions between cohort and height loss on total and cause-specific mortality were examined by inclusion of the corresponding product term in models of linear and dichotomised height loss.

All statistical analyses were performed in SAS version 9.3 (SAS Institute, Cary, NC). Results with p-values less than 0.05 were considered significant (two-sided test).

### Patient and public involvement

Patients or the public were not involved in this specific research project.

### RESULTS

Characteristics of the study population are presented in table 1. At baseline, the average age of the Swedish and Danish cohorts was 47 and 44 years, respectively,

and two-thirds were aged 38–52 years at baseline. In the pooled cohort, women lost on average 0.8 cm of height (range −1.9 to 14.0) over 11.4 years. During follow-up over a maximum of 19.3 years, 316 and 309 cases of death occurred in the Swedish and Danish cohorts, respectively. CVD was the primary cause in 157 cases, including 37 cases of stroke, while 362 cases were due to non-CVD causes (in total 519 deaths).

The first stage of the analyses examined mortality irrespective of cause. The risk for total mortality for each cm of height loss was 1.18 (1.09 to 1.28) for the Swedish women and 1.24 (1.13 to 1.37) for the Danish women, after adjusting for age, baseline height and time between height measures (table 2). Further adjustment for baseline weight and lifestyle factors gave similar estimates. Major height loss, defined as height loss greater than 2 cm, was associated with 74% and 80% higher hazard

**Table 2** Hazard ratios* and 95% CIs for association between height loss and total mortality and cause-specific mortality in Swedish and Danish women (n=2406)

| Sample | Total mortality | | | | | |
| --- | --- | --- | --- | --- | --- | --- |
| | Model 1† | | | Model 2‡ | | |
| | Swedish | Danish | Pooled | Swedish | Danish | Pooled |
| No. of cases/censored* | 316/831 | 309/950 | 625/1781 | 314/821 | 292/893 | 606/1714 |
| Height loss (cm) | 1.18 (1.09 to 1.28) | 1.24 (1.13 to 1.37) | 1.20 (1.13 to 1.27) | 1.14 (1.05 to 1.23) | 1.21 (1.09 to 1.35) | 1.15 (1.09 to 1.23) |
| Height loss, binary | | | | | | |
| Stable (≤2 cm) | 1 (reference) | 1 (reference) | 1 (reference) | 1 (reference) | 1 (reference) | 1 (reference) |
| Major height loss (>2 cm) | 1.86 (1.41 to 2.44) | 1.90 (1.36 to 2.64) | 1.85 (1.50 to 2.29) | 1.74 (1.32 to 2.29) | 1.80 (1.27 to 2.54) | 1.74 (1.41 to 2.16) |

| Pooled sample | Total CVD mortality | | Stroke mortality | | Non-CVD mortality | |
| --- | --- | --- | --- | --- | --- | --- |
| | Model 1† | Model 2‡ | Model 1† | Model 2‡ | Model 1† | Model 2‡ |
| No. of cases/censored* | 157/2249 | 156/2164 | 37/2369 | 36/2284 | 362/2044 | 347/1973 |
| Height loss (cm) | 1.28 (1.17 to 1.40) | 1.21 (1.10 to 1.32) | 1.36 (1.15 to 1.61) | 1.30 (1.09 to 1.55) | 1.18 (1.09 to 1.28) | 1.14 (1.05 to 1.24) |
| Height loss, binary | | | | | | |
| Stable (≤2 cm) | 1 (reference) | 1 (reference) | 1 (reference) | 1 (reference) | 1 (reference) | 1 (reference) |
| Major height loss (>2 cm) | 2.44 (1.69 to 3.53) | 2.14 (1.47 to 3.12) | 2.81 (1.38 to 5.75) | 2.31 (1.09 to 4.87) | 1.78 (1.34 to 2.36) | 1.71 (1.28 to 2.29) |

*Number of deaths since second height measure/censored at end of follow-up.
†Model 1 adjusted for age at follow-up, age², time interval and height at baseline. Analyses of pooled sample also adjusted for cohort (Swedish/Danish).
‡Model 2 adjusted for age at follow-up, age², time interval, height at baseline, weight at baseline, baseline smoking status, alcohol intake at baseline, baseline education and leisure time physical activity at baseline and follow-up. Analyses of pooled sample also adjusted for cohort (Swedish/Danish).

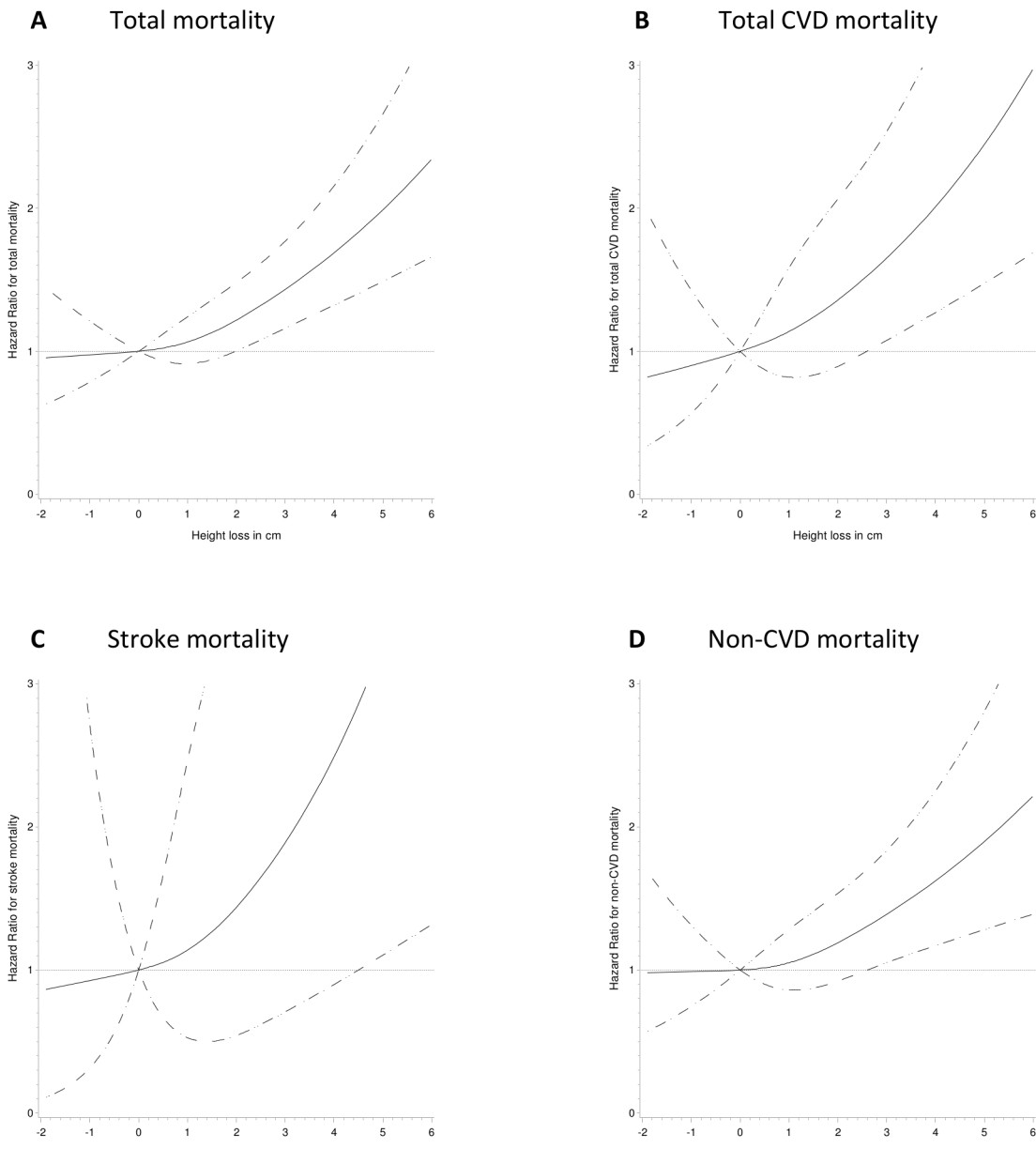

**Figure 1** Hazard ratio (HR) for 10–13 year decrease in height in relation to total mortality and cause-specific mortality in a pooled sample of Swedish and Danish women. Models were adjusted for age at follow-up, age², time interval between height measures, height at baseline, weight at baseline, baseline smoking status, alcohol intake at baseline, baseline education, and leisure time physical activity at baseline and follow-up. Zero change in height was used as reference value for HR. Solid lines represent HR and dotted lines represent 95% CI of HR. (A) Total mortality; test for curvature p=0.3, test for overall significance of curve p=0.0001, test for linearity p<0.0001. (B) CVD mortality test for curvature p=0.7, test for overall significance of curve p=0.002, test for linearity p=0.0005. (C) Stroke mortality. Test for curvature p=0.7, test for overall significance of curve p=0.06, test for linearity p=0.02. (D) Non-CVD mortality. Test for curvature p=0.4, test for overall significance of curve p=0.01, test for linearity p=0.004.

for total mortality in Swedish and Danish women, respectively. Pooled analyses suggested that height loss was associated with a monotonic increase in total mortality risk (figure 1) with a significant association with height loss of more than 2 cm. Inclusion of interaction terms between cohort and height loss showed no sign of spatial heterogeneity (p=0.5 for linear height loss, p=1.0 for major height loss). Age (years) as linear factor was associated with total mortality in pooled analyses (HR 1.12 (95% CI 1.11 to 1.14); data not shown). Stratification by baseline

age showed that the effect was similar in women aged less than 50 and those aged 50 years or older at baseline in model with continuous height (interaction p=0.9) as well as dichotomised height (interaction p=0.6) (online supplemental table S1).

All further analyses on cause-specific mortality were performed on the pooled sample due to the limited number of outcomes. These analyses suggested that continuous height loss was associated with total CVD mortality, stroke mortality and non-CVD mortality (table 2). Major

height loss was associated with a HR (95% CI) of 2.31 (1.09 to 4.87) for stroke mortality, 2.14 (1.47 to 3.12) for total CVD mortality and 1.71 (1.28 to 2.29) for mortality due to other causes, independent of age, time interval between height measures, baseline height, baseline weight, cohort and lifestyle factors (table 2). However, stroke mortality did not account for the effect of height loss on overall mortality because height loss was also associated with non-stroke CVD mortality (HR 1.82 (95% CI 1.43 to 2.32); data not shown). No significant interactions were found between height loss and cohort on cause-specific mortality (p-values for interaction >0.20 for all cause-specific mortality outcomes).

Sensitivity analyses were performed to rule out the possibility that the continuous effect of decreased height on overall mortality was driven by large decreases in only a few individuals. Excluding women with a loss of height of at least 5 cm (n=11) did not weaken the effect estimate of linear height on total mortality: HR 1.21 (95% CI 1.10 to 1.34). Similarly, when further excluding 14 women with a loss of at least 4 cm, the effect estimate remained at the same magnitude: 1.19 (1.08 to 1.32). Excluding deaths occurring during the first 2 years of follow-up (21 cases of total mortality, 6 cases of CVD mortality and 15 cases of non-CVD mortality) did not alter the results (data not shown).

Age-adjusted associations between covariates and height loss were also investigated (data not shown). High baseline height was consistently associated with greater subsequent height loss, both linear height loss (β=0.007, p=0.03) and major height loss (OR 1.03 (95% CI 1.00 to 1.06)). Aside from baseline height, the only other factor associated with height loss was baseline LTPA. High baseline LTPA was, compared with medium baseline LTPA, linearly associated with lower height loss (β=−0.15, p=0.02) while the association with major height loss was not significant (OR 0.85 (95% CI 0.52 to 1.37)). When comparing low versus medium LTPA at baseline, no associations were observed with height loss (data not shown).

## DISCUSSION
### Statement of principal findings
In this study of middle-aged Nordic women, major height loss was associated with an increased hazard of overall mortality of around 80%. Specifically, major height loss was associated with CVD mortality, with more than a twofold risk for stroke mortality. The findings were independent of age, time between height measures, cohort, and baseline values of height, weight, education and lifestyle factors.

### Strengths and limitations of the study
The main strengths of this study are its population-based sampling, the prospective designs, the standardised measurement protocols and the long durations of follow-up for endpoints. Furthermore, trained staff measured height using standardised methodology, and

mortality was ascertained through high-quality national registries,[17 18] thereby limiting potential bias in both exposure and outcome. Also, differences between the cohorts in study design, age and period were minor with results for overall mortality that were strikingly similar in both samples, and results for CVD mortality that were independent of time between height measures, age at follow-up and cohort. However, we acknowledge the risk for bias due to non-participation in both cohorts. We performed both stratified and pooled analyses, and investigated interactions by cohort, with consistent results in both cohorts. The consistency of cohort-specific results may indicate that non-participation has a minor influence on the association analyses. Another limitation is that deaths due to stroke were quite few, which implies that these results should be interpreted with some caution. Additionally, based on the observational design we cannot rule out residual confounding by unmeasured factors, such as early life exposure of physical activity and smoking, peak bone mass, diseases and medical treatments. Based on the fact that bone health is different depending on ethnicity it is important to take this factor into account when interpreting the results and generalising the results. In MONICA, subjects with non-Danish origin were not included,[14] and in the Swedish cohort only 0.3% were born outside Europe.[13] Hence, it can be concluded that the vast majority of participants were Caucasians.

### Comparison with other studies and interpretation
To the best of our knowledge, this study is the first to report the results on the effect of height loss on mortality in women followed from middle age. Previous studies of men and mixed samples, including populations followed from a baseline age of 40 years, have shown height loss to be associated with total mortality, with risk ratios or HRs between 1.45 and 3.43.[9–11] Previous studies of female populations were started at older ages and their results are discordant. Hillier and colleagues showed that in a cohort of women aged 65 years or older at baseline, losing at least 5 cm over 15 years was associated with an increased hazard for mortality of 45% compared with women losing less height.[12] However, another study of women of the same baseline age found no significant effect of 4 year height loss of more than 2 cm.[9] The lack of association in that study could possibly be explained by low power due to few women classified as exposed; additionally the low number of deaths during follow-up. Compared with the above-mentioned studies, the present study shows overall mortality estimates within the range of those previously reported, indicating a consistent association between height loss and mortality in both women and men, and over the adult life course. The latter was further confirmed by the congruent results in women younger than 50 years and those aged 50 years or older in the present study.

Moreover, the present study points out that height loss specifically predicts CVD mortality, whereas results from previous studies are somewhat divergent. Wannamethee

and colleagues[10] found an increased risk of CVD mortality in men. This was confirmed by Masunari and colleagues,[11] who presented results for women and men together. On the contrary, Auyeng and colleagues[9] found no association in either men or women, but yet again, the number of people exposed and the number of cases in this study were low. To our knowledge, the novel finding in the present study showing a particular association between height loss and stroke mortality has not been reported before. Wannamethee and colleagues[10] investigated height loss in relation to incident major stroke events in men, but found no such association. Furthermore, height loss and CVD are linked, both epidemiologically and mechanistically, by interrelations between bone loss or osteoporosis and CVD. A recent systematic review and meta-analysis found that low baseline bone mineral density (BMD) and fractures were associated with an increased risk of developing CVD.[19] The pathophysiological links between these conditions are not fully understood but presumably involve chronic inflammation and oxidative stress.[20] Additionally, similarities exist in the process of bone formation and vascular calcification, including involvement of a range of bone biomarkers.[20] Frailty, a clinical syndrome defined by impaired physical resources,[21] is another feature linked to osteoporosis[22] and CVD.[23] Different definitions of frailty exist but one of the most commonly used is the one operationalised by Fried and colleagues[21] in which frailty is defined by the occurrence of three out of five criteria: unintentional weight loss, self-reported exhaustion, weakness, slow walking speed, and low physical activity, but not height loss. Weakness has been shown as a predominant initial sign of frailty.[24] Weakness could be attributed to sarcopenia, a muscle disease common in older adults diagnosed by low muscle function in the presence of low muscle quantity or quality.[25] Sarcopenia is an age-related process, but can also stem from inflammation, malnutrition, and physical inactivity.[25] Thus, height loss is not recognised in the definition of frailty but it should be highlighted that height loss shares both the feature of impaired physical resources and the aetiology for sarcopenia and therefore future frailty definitions could consider height loss as a potentially important criterion.

## Implications of the findings

The results from the present study may be generalised to northern latitude Caucasian women. Within the northern latitude, British men have previously been investigated.[10] Populations within the northern latitude are over-represented when it comes to osteoporotic fractures[8] and, although we have not been able to find publications on differences of height loss across latitudes, it could be hypothesised that populations within the northern latitude lose more height when ageing. To gain further understanding, more studies of women and men from this region are warranted to improve knowledge about the relation between height loss, morbidity and mortality.

Height is a simple measure that could be taken in every clinical setting compared with, for example, measurement of BMD, which requires advanced methodology. Despite its simplicity, height measurement is rarely included in the clinical examination by a general practitioner. Taken together, these results suggest that height loss should be recognised within primary care to facilitate actions for CVD prevention, but others also indicate height loss as an important indicator of low BMD, vertebral fractures and vitamin D deficiency.[6]

Knowledge on how to prevent height loss is sparse. Pharmaceutical treatment for osteoporosis with alendronate has shown to prevent height loss in addition to improving bone mineral content,[26] while supplements with calcium and vitamin D have not been proven to prevent height loss.[27] Concerning lifestyle, physical activity has been identified as protective against height loss in postmenopausal women.[28] Our results confirmed that regular physical exercise could contribute significantly to height loss prevention. Still, these results suggest that moderate activity may not be enough to prevent height loss and only one in seven women in the current cohorts were active enough to benefit from physical activity in relation to decreased height loss. More research is thus needed, not only on the consequences of height loss but also on the causes to facilitate prevention of height loss and associated comorbidity and mortality.

## CONCLUSIONS

Height loss during mid-life is a risk marker for earlier mortality in northern European women. Specifically the hazard of CVD mortality is increased in women with height loss, and the results suggested that stroke mortality may be a major contributor to the total CVD association. These findings suggest the need for increased attention to height loss to identify individuals at increased CVD risk. Moreover, regular physical activity may be beneficial not only in prevention of CVD, but also in prevention of height loss and thereby further contributing to CVD prevention.

**Author affiliations**

[1]School of Public Health and Community Medicine, Institute of Medicine, Sahlgrenska Academy, University of Gothenburg, Gothenburg, Sweden
[2]Department of Internal Medicine and Clinical Nutrition, Institute of Medicine, Sahlgrenska Academy, University of Gothenburg, Gothenburg, Sweden
[3]The Parker Institute, and the Center for Clinical Research and Prevention, Bispebjerg and Fredriksberg Hospitals, The Capital Region, Denmark
[4]Primary Health Care/School of Public Health and Community Medicine, Institute of Medicine, Sahlgrenska Academy, University of Gothenburg, Gothenburg, Sweden
[5]Section for General Practice, Department of Public Health, University of Copenhagen, Copenhagen, Denmark

**Contributors** BLH and LL initiated the project. SK and KM performed all statistical analyses. SK had main responsibility for writing the article. SK, KM, RD, CB, BLH and LL all contributed to the statistical analyses and interpretation and provided comments on the manuscript. SK, KM, RD, CB, BLH and LL all read and approved the final manuscript. BLH and LL shares last authorship.

**Funding** This work was supported by grants from the Swedish Research Council (VR521-2010-2984), the Swedish Research Council for Health, Working Life, and Welfare (2006–1506), and the Swedish ALF-agreement (ALFGBG-672971 and ALFGBG-722441).

**Disclaimer** The funding sources had no role in study design, collection, analysis and interpretation of data, the writing of the report, or the decision to submit the article for publication.

**Competing interests** None declared.

**Patient consent for publication** Not required.

**Ethics approval** Participants from both study cohorts provided informed consent to participate. Since 1980, all examinations in PPSWG have been approved by the Regional Ethics Review Board in Gothenburg, in accordance with the Declaration of Helsinki (registration number T331-14 for linkage to national registries). The MONICA study was approved by the Local Ethics Committee of Copenhagen County and the Danish Data protection Office (J.nr 2015-41-3942), and was in accordance with the principles of the Helsinki Declaration.

**Provenance and peer review** Not commissioned; externally peer reviewed.

**Data availability statement** Data underlying this article are available upon reasonable request. Data cannot be made publicly available for ethical and legal reasons. Public availability may compromise participant privacy, and this would not comply with Danish or Swedish legislation. Requests for data should be addressed to Professor Berit L Heitmann (Berit.Lilienthal.Heitmann@regionh.dk) and Professor Lauren Lissner (lauren.lissner@gu.se) who will provide the data access in accordance with the Danish and Swedish Data Protection Agency, respectively.

**ORCID iDs**
Sofia Klingberg http://orcid.org/0000-0002-9093-2826
Cecilia Björkelund http://orcid.org/0000-0003-4083-7342
Berit Lilienthal Heitmann http://orcid.org/0000-0002-6809-4504

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
