## [Reviewer comments · BMJ Open]

ARTICLE DETAILS

TITLE (PROVISIONAL)	Loss of height in relation to total and cardiovascular mortality: a cohort study of northern European women
AUTHORS	Klingberg, Sofia; Mehlig, Kirsten; Dangol, Rojina; Björkelund, Cecilia; Heitmann, Berit; Lissner, Lauren

VERSION 1 – REVIEW

REVIEWER	Peter, Raphael Institute of Epidemiology and Medical Biometry, Ulm University
REVIEW RETURNED	07-Feb-2021

GENERAL COMMENTS	The present manuscript, „Loss of height in relation to total and cardiovascular mortality: a cohort study of northern European women”, is well written and technically sound. The conclusions drawn follow from the data presented. I only have very few minor points the authors may want to consider: 1. Hight measurement is described on page 7, lines 24-27. Please add information on the resolution of hight-measurements. Was height recorded to the nearest cm, 0.5 cm, or even mm?2. Page 10, line 3, “All statistical analyses were performed in SAS, version 9.3 (SAS institute, Cary, NC)”. “institute” may be a proper noun in this context and should be capitalized.
---

REVIEWER	Wang, Jinfeng Institute of Geographic Sciences and Natural Resources Research, Chinese Academy of Sciences
REVIEW RETURNED	20-Feb-2021

GENERAL COMMENTS	1. Explain “representative sample of 1462”. A sample is biased if the population is stratified heterogeneity and not all strata are sampled.2. The time lag as a predictor?3. Compare the prediction accuracy: age and height as two predictors, respectively.4. Measure and attribute the spatial stratified heterogeneity of the outcomes between the two counties.
--

REVIEWER	Sung, Jidong Sungkyunkwan University School of Medicine, Division of Cardiology, Department of Medicine
REVIEW RETURNED	25-Feb-2021

GENERAL COMMENTS	This study investigated the association of height loss and mortality in in population-based samples of middle-aged Nordic women. As
---

	the authors commented in the Introduction, the study finding is not entirely new but this study was done with female cohorts providing gender-specific data, which is a significant strength. Several points should be addressed before publication.  1. How was physical activity estimated? Was it measured by a standardized tool or self-assessed response to 'custom-made' question? Variables related to cardiorespiratory fitness probably may be the strongest confounder, so accuracy of measurement should be discussed. And was it measured once at baseline? Was change of LTPA over time not considered? 2. Were study subjects free of cardiovascular diseases at the enrollment? Or was baseline health status unknown? 3. The authors seemed to assume that study subjects largely being Caucasian but data on ethnic composition was not specifically shown. 4. Discussion on the study imitation was not presented.
--	--

VERSION 1 – AUTHOR RESPONSE

Reviewer: 1

Dr. Raphael Peter, Institute of Epidemiology and Medical Biometry, Ulm University

Comments to the Author:

The present manuscript, „Loss of height in relation to total and cardiovascular mortality: a cohort study of northern European women”, is well written and technically sound. The conclusions drawn follow from the data presented.

I only have very few minor points the authors may want to consider:

1. Height measurement is described on page 7, lines 24-27. Please add information on the resolution of height-measurements. Was height recorded to the nearest cm, 0.5 cm, or even mm?

Response: Thank you for observing this. The information has been added and now reads “At both examinations, height was measured with a stadiometer to the nearest 0.5 cm on subjects without shoes.” (see page 7).

2. Page 10, line 3, “All statistical analyses were performed in SAS, version 9.3 (SAS institute, Cary, NC)”. “institute” may be a proper noun in this context and should be capitalized.

Response: This has now been corrected (see page 10).

Reviewer: 2

Dr. Jinfeng Wang, Institute of Geographic Sciences and Natural Resources Research

Comments to the Author:

1. Explain “representative sample of 1462”. A sample is biased if the population is stratified heterogeneity and not all strata are sampled.

Response: We agree that this wording was not correct and have now clarified the sampling

procedure. The description of the sampling procedure now reads “The PPSWG, initiated in 1968-1969, recruited a sample of 1462 women, born in 1908, 1914, 1918, 1922 and 1930 on specific dates that were distributed evenly across birth years. Over 90% of the invited women participated at the baseline health examination (13).” (see page 6).

2. The time lag as a predictor?

Response: In both cohorts, time between height examinations could differ with observation time ranging between 9.8 and 13.4 years. To account for differences in time for exposure to occur, this variable was included as a covariate in all analyses of height loss in relation to mortality.

3. Compare the prediction accuracy: age and height as two predictors, respectively.

Response: Although none of the age associations are shown in our tables, we agree that age is a powerful predictor of mortality. Therefore, all models were adjusted for age at start of follow up at the second examination of height. At that time, the age was ranging from 40 to 70. To point out the importance of age as a predictor of mortality we have now included this in the results: “Age (year) as linear factor was associated with total mortality in pooled analyses (HR (95% CI) 1.12 (1.11 to 1.14)) (data not shown).” (see page 11).

4. Measure and attribute the spatial stratified heterogeneity of the outcomes between the two counties.

Response: Stratified analyses suggested no spatial heterogeneity, as evidenced by effect estimates for total mortality that were very similar in both cohorts (see table 2). In analyses of cause specific mortality we pooled the cohorts in order to increase the number of cases, and these were controlled for survey country. Based on the comment from the reviewer, we investigated interactions between country and height loss on the effect of cause-specific mortality, but found no significant interactions. This has now been added to the methods on page 10 and to the results on page 12.

Reviewer: 3

Dr. Jidong Sung, Sungkyunkwan University School of Medicine

Comments to the Author:

This study investigated the association of height loss and mortality in in population-based samples of middle-aged Nordic women. As the authors commented in the Introduction, the study finding is not entirely new but this study was done with female cohorts providing gender-specific data, which is a significant strength. Several points should be addressed before publication.

1. How was physical activity estimated? Was it measured by a standardized tool or self-assessed response to ‘custom-made’ question? Variables related to cardiorespiratory fitness probably may be the strongest confounder, so accuracy of measurement should be discussed. And was it measured once at baseline? Was change of LTPA over time not considered?

Response: Leisure time physical activity (LTPA) was self-assessed by a question with four response alternatives ranging from sedentary to vigorous activity. This method, the “Saltin–Grimby Physical Activity Level Scale”, has been widely used in Scandinavia and validated against objective measures such as maximal oxygen uptake. Information about the method and its application for health research has been compiled by Grimby et al. 2015 (reference 15 now included, see page 7). LTPA was assessed with the same questionnaire at baseline and follow-up, and LTPA at follow-up has now been included in model 2 (see Table 2, Table S1, Figure 1), and the manuscript has been updated accordingly. Inclusion of LTPA at follow-up did only have minor impact on the effect estimates.

Information about LTPA at follow-up examination has also been included in the descriptive table (see Table 1).

2. Were study subjects free of cardiovascular diseases at the enrollment? Or was baseline health status unknown?

Response: Study subjects with cardiovascular disease were not excluded from the analyses. This decision was based on the fact that we investigated mortality and not incident CVD. Therefore, some of our cause specific mortality cases included participants with prevalent CVD at baseline. A sensitivity analyses excluding cases within the first two years of follow up for the respective outcome has now been added (see page 9 and 12). Exclusion of cases within the first two years of follow up did not alter the results.

3. The authors seemed to assume that study subjects largely being Caucasian but data on ethnic composition was not specifically shown.

Response: Thank you for addressing this point. We have now raised this in the discussion (see page 14): “Based on the fact that bone health is different depending on ethnicity it is important to take this factor into account when interpreting the results and generalizing the results. In Monica, subjects with non-Danish origin were not included (14), and in the Swedish cohort only 0.3 % were born outside Europe (13). Hence, it can be concluded that the vast majority of participants were Caucasians.”.

4. Discussion on the study imitation was not presented.

Response: In the initial manuscript, study limitations regarding number of cases and residual confounding were discussed. In the revised manuscript a discussion on ethnicity has been added (see answer above and page 14).

VERSION 2 – REVIEW

REVIEWER	Wang, Jinfeng Institute of Geographic Sciences and Natural Resources Research, Chinese Academy of Sciences
REVIEW RETURNED	23-Apr-2021

GENERAL COMMENTS	1. Please note that a big sample must not guarantee the unbiasedness of the sample; 2. Spatial stratified heterogeneity of the outcomes between the two counties should be measured and attributed by strict statistics, rather than by “looking”.
---

REVIEWER	Sung, Jidong Sungkyunkwan University School of Medicine, Division of Cardiology, Department of Medicine
REVIEW RETURNED	25-Apr-2021

GENERAL COMMENTS	The issues raised by the reviewer have been adequately addressed and the manuscript has been improved. Thank you for the effort.
--

VERSION 2 – AUTHOR RESPONSE

Reviewer: 2

Dr. Jinfeng Wang, Institute of Geographic Sciences and Natural Resources Research Comments to the Author:

1. Please note that a big sample must not guarantee the unbiasedness of the sample;

Response: Thank you for this comment. We agree and have now clarified the risk of bias on page 14: *“However, we acknowledge the risk for bias due to non-participation in both cohorts. We performed both stratified and pooled analyses, and investigated interactions by cohort, with consistent results in both cohorts. The consistency of cohort-specific results may indicate that non-participation has a minor influence on the association analyses.”* .

2. Spatial stratified heterogeneity of the outcomes between the two counties should be measured and attributed by strict statistics, rather than by “looking”.

Response: Thank you for addressing this. Interaction analyses between cohort and height loss on cause-specific mortality was briefly addressed on page 12 and we have now also added p-values for the interaction: *“No significant interactions were found between height loss and cohort on cause-specific mortality (p-values for interaction >0.20 for all cause-specific mortality outcomes).”*. In addition, we have also included interaction analyses between cohort and height loss on total mortality and these showed no sign of spatial heterogeneity. The following text has been added to page 11: *“Inclusion of interaction terms between cohort and height loss in pooled analyses showed no sign of spatial heterogeneity (p-values for interaction >0.50 for total mortality).”*.

Reviewer: 3

Dr. Jidong Sung, Sungkyunkwan University School of Medicine

Comments to the Author:

The issues raised by the reviewer have been adequately addressed and the manuscript has been improved. Thank you for the effort.

Response: Thank you very much.